# Radiologists’ Communicative Role in Breast Cancer Patient Management: Beyond Diagnosis

**DOI:** 10.3390/healthcare12111145

**Published:** 2024-06-05

**Authors:** Luciano Mariano, Luca Nicosia, Adriana Sorce, Filippo Pesapane, Veronica Coppini, Roberto Grasso, Dario Monzani, Gabriella Pravettoni, Giovanni Mauri, Massimo Venturini, Maria Pizzamiglio, Enrico Cassano

**Affiliations:** 1Breast Imaging Division, IEO, European Institute of Oncology IRCCS, Via Ripamonti, 435, 20141 Milan, Italy; luciano.mariano@ieo.it (L.M.); filippo.pesapane@ieo.it (F.P.); maria.pizzamiglio@ieo.it (M.P.); enrico.cassano@ieo.it (E.C.); 2Department of Biotechnology and Life Sciences, University of Insubria, Via J.H. Dunant, 3, 21100 Varese, Italy; 3Postgraduation School in Radiodiagnostics, Faculty of Medicine and Surgery, University of Milan, 20122 Milan, Italy; adriana.sorce@unimi.it; 4Applied Research Division for Cognitive and Psychological Science, IEO, European Institute of Oncology IRCCS, 20141 Milan, Italy; veronica.coppini@ieo.it (V.C.); roberto.grasso@ieo.it (R.G.); dario.monzani@ieo.it (D.M.); gabriella.pravettoni@ieo.it (G.P.); 5Department of Oncology and Hemato-Oncology, University of Milan, 20122 Milan, Italy; 6Laboratory of Behavioral Observation and Research on Human Development, Department of Psychology, Educational Science and Human Movement, University of Palermo, 90128 Palermo, Italy; 7Division of Interventional Radiology, IEO, European Institute of Oncology IRCCS, Via Ripamonti, 435, 20141 Milan, Italy; giovanni.mauri@ieo.it; 8Diagnostic and Interventional Radiology Unit, ASST Settelaghi, Insubria University, 21100 Varese, Italy; massimo.venturini@uninsubria.it

**Keywords:** breast cancer, communication, psychological support, doctor–patient relationship

## Abstract

In the landscape of cancer treatment, particularly in the realm of breast cancer management, effective communication emerges as a pivotal factor influencing patient outcomes. This article delves into the nuanced intricacies of communication skills, specifically spotlighting the strategies embraced by breast radiologists. By examining the ramifications of communication on patient experience, interdisciplinary collaboration, and legal ramifications, this study underscores the paramount importance of empathetic and comprehensive communication approaches. A special emphasis is placed on the utilization of the SPIKES protocol, a structured method for conveying sensitive health information, and the deployment of strategies for navigating challenging conversations. Furthermore, the work encompasses the significance of communication with caregivers, the integration of artificial intelligence, and the acknowledgement of patients’ psychological needs. By adopting empathetic communication methodologies and fostering multidisciplinary collaboration, healthcare practitioners have the potential to enhance patient satisfaction, promote treatment adherence, and augment the overall outcomes within breast cancer diagnosis. This paper advocates for the implementation of guidelines pertaining to psychological support and the allocation of sufficient resources to ensure the provision of holistic and patient-centered cancer care. The article stresses the need for a holistic approach that addresses patients’ emotional and psychological well-being alongside medical treatment. Through thoughtful and empathetic communication practices, healthcare providers can profoundly impact patient experiences and breast cancer journeys in a positive manner.

## 1. Introduction

In recent years, there has been a notable shift in the landscape of cancer care, with an increasing emphasis on the need and ability to provide patients with clear and comprehensive information [1]. This change is supported by recognizing the pivotal role of a thorough understanding of diagnosis and treatment in the healing process [1]. Several studies confirm how healthcare professionals’ communication skills positively influence patients’ quality of life, therapeutic plan adherence, and illness progression and outcome [2]. Furthermore, providing patients with detailed and easily accessible information demystifies the medical pathway, enabling them to face the disease with a greater confidence and peace of mind [3,4].

This clinical–scientific evidence is particularly pronounced in managing breast cancer (BC) patients [2]. The relation with this disease often exerts a negative influence on patients’ psychosocial well-being, which can be significantly alleviated through a doctor–patient relationship based on trust that addresses their needs and provides essential psychological support, an essential prerequisite for forming robust diagnostic–therapeutic compliance and supports the patients through one of the most challenging experiences of their lives.

In this context, breast radiologists (BR), key figures in BC prevention, diagnosis, and follow-up programs, routinely navigate complex communicative scenarios. Their role entails a crucial responsibility to establish a proper relationship with the patient and caregiver, especially when revealing the disease’s initial diagnosis or progression [5,6]. The BR’s ability to articulate results, discuss potential diagnoses, and guide patients through the most appropriate diagnostic and therapeutic pathway is essential to ensure comprehensive patient-centered care. These communication skills enhance the overall patient experience and strengthen interdisciplinary collaboration, emphasizing the radiologist’s role as an interpreter of images and as a compassionate and communicative teammate in the pursuit of health and well-being. As a result, it is imperative to enhance the physicians’ training in communication skills throughout their education [7]; this proficiency is likewise linked to decreased legal complications and improved patient outcomes [7].

Studies suggest that communication skills do not significantly improve with experience, hence considerable efforts are invested in developing training courses that can enhance the communication skills of professionals working with oncology patients. For instance, Moore et al. conducted an excellent review analyzing 15 trials carried out in Europe, aiming to assess whether CST was effective in improving communication skills, patient health status, and satisfaction [8].

What they concluded is that some courses are effective in improving certain communication skills related to information gathering and supportive skills, although it is still not clear which type, duration, and intensity of training are most effective and whether consolidation workshops can enhance the impact of the training [8].

This article briefly overviews the essential physician communication skills in BC patient management, focusing on the BR approach and emphasizing their impact on patient experience, interdisciplinary collaboration, and legal aspects. Communication with the patient’s caregiver is also explored, highlighting the delicacy of involvement in the diagnostic process, as well as the possible role of Artificial Intelligence (AI) in optimizing doctor–patient communication, considering the potential of decision support system based on advanced algorithms in managing clinical information and facilitating communication between healthcare professionals and patients. The aim is to provide comprehensive support to patients, their families, and partners while facing the challenges of a BC journey.

## 2. Communication Skills in Breast Cancer

The skills that are the subject of patient communication training are varied and often described disparately [9].

Doctors are encouraged to provide as much information as possible to offer therapeutic choices and discuss their emotional implications.

Complementarily, patients who perceive themselves as more competent and involved in communication with healthcare providers tend to derive greater satisfaction from such interactions [10].

Some courses specifically designed to enhance communication skills have demonstrated effectiveness, and CST based on acquiring skills seems to be more effective than programs based on attitudes or specific tasks [3,7,11].

Moore et al., in their review, reported some interesting analyses about the effectiveness of communication skills training in changing behavior of HCPs working in cancer care and in improving patient health status and satisfaction [8].

In particular, Gibon et al. in 2011 [12] and Kruijver at al. in 2001 [13] reported a significantly greater emotional depth in the intervention groups compared with the control groups [8].

In 2011, Fujimori et al. [3] reported an improvement in emotional support scores in the intervention group compared with the control group [8], while Wilkinson et al. in 2008 [14] and Golez et al. in 2009 [15] reported significantly better global communication scores for the CST groups than the control groups.

Various communicative models have also been developed to facilitate the exchange of information between healthcare providers and patients. These models aim to create a “mental map” that captures all the crucial elements necessary to optimize conversations and mitigate their emotional impact. Additionally, courses designed to improve communication skills emphasize the importance of incorporating authentic patient communication experiences as robust learning tools [16,17,18,19].

Even more so, role-playing with simulated patients trained to portray individuals with cancer proves to be an effective training method for practice without causing distress to actual patients [8].

An example of a widely accredited protocol in the scientific field is SPIKES (Figure 1), which is used to communicate negative health news to patients [19,20]. This protocol, structured into six phases, aims to achieve four main objectives: obtaining information from the patient, transmitting medical information, supporting the patient, and involving them in developing the treatment strategy.

Two fundamental aspects of this protocol are represented by “giving information” (Knowledge) and “evaluating the patient’s emotions” (Emotions). In the first case, appropriately informing the patient about negative health news can help reduce the state of shock [21]. In transmitting medical information, it is essential to use understandable language to the patient, avoid technical jargon, and provide information in appropriate doses [21,22,23]. The BR should explain the examination procedure, potential risks, and treatment recommendations in simple and understandable words (e.g., instead of saying: “We will use a contrast agent to enhance the images”, the BR might say: “We’ll use a harmless dye to make the pictures clearer”, or instead of saying: “This lymph node looks bad and needs to be checked urgently”, one could say “We noticed a lymph node that appears slightly different than normal”).

Dealing with the patient’s emotional reaction is one of the most delicate challenges after communicating negative news. Even in radiology, effective communication is vital for managing patient emotions [24,25,26,27,28]. The doctor must pay attention to the patient’s verbal and non-verbal reactions, responding empathetically and offering support. Understanding the reasons behind the patient’s reactions can help establish a meaningful emotional connection during the interaction. The BR should demonstrate a genuine interest in the patient’s well-being and understand their concerns and fears [29]. These practices create a supportive environment, ensuring that patients feel heard and understood during their medical journey. For instance, it would be good practice to greet the patient with a warm smile, ask how they are feeling by using open-ended questions (e.g., “How are you feeling about the upcoming procedure?”), focus on psychological aspects (e.g., addressing fears and concerns), show empathy (e.g., acknowledging the patient’s emotions), and provide clear information (e.g., explaining the procedure in simple terms) [30]. Moreover, patients should be encouraged to express their concerns and questions, ensuring they respond comprehensively and courteously (e.g., you might say to the patient: “I’m here to answer all your questions and reassure you throughout the process. Feel free to ask anything that worries you”). Finally, the BR must be sensitive to the patient’s needs and sensitivities, respecting their privacy and maintaining a comfortable environment during the examination. For instance, they could offer the patient a blanket to keep warm during the examination or inquire if they have any music or room lighting preferences.

## 3. Informing the Patient

In technologically and culturally advanced societies, the requirement for the assisted individual to be informed about the prevention, diagnosis, and potential therapy of their ongoing or at-risk illness represents a principle that can be summarized as the following reasons: legal (since 1998, professional codes of ethics and regulations have mandated this) [31]; ethical (every human being has the right to information regarding all aspects of their health status and life) [32]; scientific (accurate information about the illness and resources available to address it improves the emotional adaptation to the disease) [33,34]; professional (understanding and adherence to medical prescriptions, along with satisfaction with received care, reduce medical–legal disputes and the risk of burnout for the physician) [35,36].

There is a unanimous agreement that informing the patient is the attending physician’s responsibility. Article 27 of Title III “Relationships with the Assisted Person” of the Deontological Code [37,38] clearly states that the patient’s free choice of doctor and place of treatment is an inalienable right. Any agreement between doctors that could influence this freedom of choice is prohibited, although it is permissible to recommend consultants or suitable treatment facilities. The importance of this article lies in preserving and ensuring the right of choice, one of the fundamental human rights. This right is evident in the patient and physician interaction during diagnostic and therapeutic procedures.

Case law has also marked this direction: an Italian regional law that regulates the composition of BC units, adopted as a model in Italy, requires unit members to identify a person responsible for communicating the diagnosis [39]. The presence of an experienced psychologist who can assist the involved parties in processing the emotions following critical communication can promote better psychological adjustment and alleviate the burden of stress on the physician by sharing the emotional load. Among all critical areas of medicine, the most significant contribution comes from psychologists working at Oncology Institutes [40]. The presence of a psychologist becomes even more important in situations where new genetic-based diagnostic methodologies identify individuals susceptible to neoplastic pathology, such as in the case of BRCA-1 and -2 genes associated with breast and ovarian tumors [41]. Indeed, as genetic testing becomes more and more prevalent, it is ethically necessary to ensure that the benefits of such testing outweigh potential harm, particularly concerning psychological distress [26,42,43]. Research indicates that while genetic testing may uncover crucial information, it can also potentially cause adverse psychological reactions in those who are found to be carriers; feelings of overwhelmedness, anxiety, and depression can occur, as well as important distress both in patients and caregivers [44,45]. On a more positive note, the literature also reports that while carriers may experience increased emotional distress, it typically returns to baseline levels within a year [46]. In addition, it is vital to consider that factors such as age, family history, and communication dynamics influence psychological outcomes and may necessitate additional support during counseling [47,48].

Moreover, presenting distressing information to patients and family members in an organized and appropriate manner is never simple. However, providers can refer to available guidelines, such as the SPIKES protocol, that could help them manage the relationship with the patients and their loved ones when negative or tough news is to be shared. SPIKES is an acronym for presenting distressing information to patients and families in an organized manner, and it provides suggestions that can help healthcare providers whether there is no access to the mediation of a psycho-oncologist [49].

The step of giving information is usually quite delicate and starts with asking the patients and family members how much and what kind of information will be helpful or desired. As a matter of fact, not every patient wants to have complete knowledge of the risks or complications, and the medical team is obliged to respect these types of needs [50].

Breast radiologists are pivotal figures in the journey of breast cancer patients, primarily due to their adept communication skills. As the first point of contact to relay the diagnosis, their capacity to convey intricate medical information with clarity and compassion significantly aids patients in navigating diagnosis, treatment decisions, and post-treatment care. Initially, during the diagnostic phase, transparent communication is paramount for patients to comprehend the results of imaging tests such as mammograms, ultrasounds, and MRIs. Breast radiologists analyze these images and relay their findings to patients, offering a comprehensive understanding of their condition and easing any apprehension or uncertainty. Furthermore, in treatment planning, radiologists collaborate closely with oncologists and surgeons to ensure that patients are fully apprised of their options and potential outcomes. They delve into the benefits and risks of various treatment modalities, empowering patients to make informed decisions aligned with their preferences and values. Moreover, breast radiologists assume a crucial role in post-treatment follow-up, diligently monitoring patients for signs of recurrence or complications. Through continual communication, they provide ongoing support and reassurance, adeptly addressing any concerns or queries that may arise. In essence, the communication prowess of breast radiologists is indispensable in delivering holistic care to breast cancer patients, fostering trust and empowerment, and ultimately leading to improved outcomes throughout their journey.

## 4. How to Set up the Conversation

In organizing a meeting aimed at delivering potentially upsetting news, such as a worsening of conditions or a BC diagnosis, several considerations are necessary to ensure effective communication and support for patients and caregivers [19]. Selecting a designated meeting area in a suitably inviting atmosphere, carefully arranged to maximize privacy and avoid interruptions, is paramount. This facilitates comfortable interaction between the BR and the patient, enabling them to converse face-to-face [51]. Pagers should be set on vibration mode, and calls should be held to maintain an optimal quiet environment for attentive and empathetic listening. Moreover, enhancing a thorough comprehension of the clinical scenario, according to the patient’s preferences, including other members from the multidisciplinary team, the patient’s partner, or a family member, can prove advantageous. This offers broader emotional support and facilitates a more comprehensive understanding and involvement in decision-making processes [51].

For instance, the BR may choose a consultation room with comfortable seating arrangements and soft lighting to create a relaxed ambiance conducive to open communication. Additionally, ensuring that the room is soundproof can help maintain confidentiality and prevent distractions. Furthermore, the BR may invite a breast care nurse or a genetic counselor to join the discussion, providing additional expertise and support to address patient concerns regarding their diagnosis or treatment options [29]. Involving other healthcare team members can offer a more holistic approach to patient care, ensuring that all aspects of the patient’s well-being are addressed [51].

Indeed, their consultations have been shown to lead to significant management changes for patients with BC, including alterations in surgical approaches and neoadjuvant chemotherapy regimens, often improving patient outcomes.

Finally, setting an appropriate atmosphere and integrating multidisciplinary expertise are crucial in facilitating difficult discussions on sensible topics. By encouraging privacy, support, caregiver participation, and collaboration among medical staff, patients can receive comprehensive care that not only addresses their medical needs and considers their mental health and emotional well-being, underscoring the importance of a patient-centered approach to managing BC [52,53].

Previous considerations are coherent with existing literature. In a recent study, Krieger and colleagues conducted semi-structured telephone interviews with adult cancer patients to inquire about their perceptions and preferences in receiving bad news and psychological support. Study results reported an important impact of the modality of communicating unfortunate information on patients’ lives, trajectories, psychosocial adjustment, and openness to psychological support [54].

While BRs have a crucial responsibility in communicating diagnosis and treatment information, it is equally important to acknowledge the limitations of their provision of comprehensive support and involvement with the psychological aspects of the patients.

Although healthcare professionals can and should learn communication techniques and follow useful guidelines such as the SPIKES protocol, the presence of a psycho-oncologist is irreplaceable, especially during delicate or emotionally complex situations involving patients and their families. Thus, multidisciplinary teamwork is always recommended to provide comprehensive care that addresses the patient’s medical and psychological aspects [55].

## 5. Technique of Conversation

As in any other clinical encounter, the BR plays a fundamental role beyond mere diagnosis when meeting BC patients. The 2023 AIOM recommendations advocate for the physician’s use of language that is entirely clear and accessible to the patient [18]. Where the use of specialized radiological terminology is indispensable, the physician should take care to explain the meaning of the terms used, never assuming that the patient has understood their significance. It is suggested to begin with open-ended questions, allowing the patient to express themselves more fully before delving into specific topics. It is also advisable to be prepared to repeat what has been communicated, as the emotional tension experienced by the patient during listening, sometimes a genuine emotional shock, can impede attentive and productive listening [18].

Additionally, utilizing certified medical interpreter services is fundamental when the BR does not speak the patient’s language. Medical interpreter use is associated with improved medical care and patient satisfaction and can potentially increase adherence to screening mammograms and follow-up in patients with limited English proficiency [56]. Moreover, it is fundamental to consider several language barriers and disparities in this context. Indeed, BC patients may present hearing impairments or disabilities that prevent them from talking or understanding spoken language, further complicating interactions with healthcare providers. These individuals may face unique communication needs that have to be addressed by providing adequate education and utilizing qualified personnel proficient in sign language or alternative communication methods. By tailoring communication strategies that meet the patients’ diverse requirements, BRs can increase the quality of standards of equitable and patient-centered care [57].

Non-verbal communication plays an equally important role [24]. Facial expressions, often more sincere than words, posture, gazes, gestures, and even clothing style, can be used as cues to understand their emotions and needs better. It is demonstrated that consistent eye contact, body posture, and the physician’s smile positively influence the understanding and retention of information received and the friendly perception of the physician’s persona [25].

In conclusion, the integration of effective verbal and non-verbal communication strategies, which also consider barriers and disparities within the population, is crucial to ensure that radiologists provide clear, useful, and tailored information to their patients when communicating adverse information.

These examples demonstrate how various communication strategies can be applied in clinical practice to improve the care and outcomes of breast cancer patients. Each approach prioritizes clear, empathetic communication and aims to empower patients to actively participate in their treatment decisions and ongoing care.

Patient-Centered Counseling: Breast cancer diagnosis can be overwhelming for patients. Effective communication strategies involve patient-centered counseling, where healthcare providers prioritize the patient’s emotional needs and preferences. For instance, providing ample time for patients to ask questions, offering emotional support, and tailoring information to their level of understanding can greatly enhance their coping mechanisms and decision-making abilities.Multidisciplinary Tumor Board Meetings: Multidisciplinary tumor board meetings bring together various specialists involved in breast cancer care, including radiologists, oncologists, surgeons, and pathologists. These meetings facilitate collaborative decision-making and ensure that all aspects of a patient’s case are thoroughly discussed. Clear communication among team members ensures comprehensive treatment planning and minimizes the risk of misunderstandings or omissions in patient care.Use of Visual Aids: Visual aids, such as diagrams, models, or educational videos, can enhance patient understanding of complex medical concepts related to breast cancer diagnosis and treatment. For example, showing patients visual representations of tumor growth, treatment procedures, and potential side effects can help them grasp information more effectively and actively participate in shared decision-making.Shared Decision-Making: Shared decision-making involves collaborative discussions between healthcare providers and patients to determine the most appropriate treatment plan based on the patient’s preferences, values, and medical circumstances. This approach empowers patients to actively participate in their care, leading to greater satisfaction with treatment decisions and improved adherence to treatment plans.Digital Health Platforms: Digital health platforms offer innovative ways to communicate with breast cancer patients and provide ongoing support throughout their journey. For example, online patient portals or mobile applications can facilitate secure messaging between patients and healthcare providers, enable access to educational resources, and track symptoms or medication adherence. These platforms promote continuous communication and engagement, enhancing the overall quality of care.

## 6. Special Considerations

Ineffective communication practices in healthcare settings encompass a range of missteps, including evading patient inquiries, disregarding their viewpoints, casting negative connotations on their expressed emotions, and prematurely suggesting psychological or psychiatric diagnoses [51]. Such practices have profound implications, potentially eroding patient morale, fostering distrust toward healthcare providers, and limiting patient autonomy, exacerbating non-cooperation.

Certainly, nonoptimal provider–patient communication could often stem from a lack of skills and reluctance to confront patients’ emotions; BRs and other medical staff may overlook the entirety of the patient’s concerns, contributing to ineffective communication.

Additionally, a cold and detached demeanor is often perceived as evasive and unsympathetic by patients, leading to frustration, anger, and feelings of anxiety and depression [58].

Physicians are urged to cultivate an environment conducive to open dialogue, fostering active patient engagement through empathetic expressions and emotional validation. This entails embracing moments of silence and acknowledging their significance in allowing patients to articulate complex thoughts and emotions.

On the other hand, offering unreasonable reassurances, belittling problems or diverting the conversation with jokes could be counterproductive strategies that may inhibit functional communication and patients’ trust in the BRs’ sincerity [59].

Furthermore, healthcare professionals should strive to eliminate biases concerning sexual orientation and gender identity, employing language that is inclusive and free from judgment.

Learning to care for members of the lesbian, gay, bisexual, transgender, queer, intersex, or asexual community (LGBTQIA+) should be implemented during the years of training and education for BRs to avoid unconscious and perceived biases, as well as inappropriate terminology and use of pronounce, unfamiliarity with gender modification surgeries and exclusive and discriminating care [60].

Effective communication is paramount in shaping patient outcomes, fostering interdisciplinary collaboration, and averting legal pitfalls within healthcare. Firstly, it is imperative for healthcare providers to engage in clear and comprehensive communication with patients to ensure that they receive optimal care and grasp their treatment plans. Miscommunication or misunderstanding can result in medication errors, overlooked diagnoses, or inappropriate treatments, all of which can detrimentally affect patient well-being and recovery. Furthermore, seamless communication among physicians, nurses, pharmacists, and other members of the care team is crucial for enhancing patient care through collaborative efforts across various specialties. This facilitates a holistic approach to patient management and facilitates the exchange of invaluable insights and expertise. Additionally, effective communication plays a vital role in mitigating legal risks in healthcare settings. Through meticulous documentation, informed consent procedures, and transparent communication with patients and their families, the likelihood of malpractice claims or legal disputes can be minimized. By ensuring that all stakeholders are well-informed and actively involved in the decision-making process, healthcare providers can mitigate the risk of misunderstandings or allegations of negligence, thus safeguarding both patient welfare and professional integrity. In summary, communication serves as a cornerstone in delivering high-quality healthcare, fostering collaboration among healthcare professionals, and preempting legal challenges in the medical realm.

## 7. Communication and Caregiver

It is common practice to involve patients’ family members, especially their partners, in planning clinical meetings. This approach is supported by evidence showing how the distress caused by illness also affects caregivers [61,62]. It is important to note that physical and psychological suffering generates conflicting feelings in the partner who oscillates between understanding the patient’s feelings and the need to manage their own negative emotions [63].

Similarly, the patient themselves is conflicted between the desire for emotional support from their partner and feelings of guilt and responsibility toward them, which sometimes lead her to seek distance.

During consultations with patient-partner pairs, particular attention should be paid to exploring the emotional and practical resources to help them cope with the illness. It is also important to value the couple’s emotional bond, support their parenting skills, and encourage mutual support to maintain a satisfactory professional and relational life [64].

Involving a professional specializing in psychological dynamics and couples counseling may be helpful. This can facilitate discussions about the couple’s sexual intimacy, often influenced by the illness. Indeed, a diagnosis of BC impacts the relationships and the sexual life of women and their partners, so it is crucial to address potential issues by discussing them with the couple, avoiding judgment, and putting them at ease [65]. However, couples often prefer to receive information via brochures or websites because of prejudice, taboos, and cultural stigma associated with sexuality, especially regarding women [66]. This holistic approach ensures that the patient and their support system receive the necessary information and emotional support during this difficult time.

## 8. Communication and Artificial Intelligence: Future Perspectives

In the medical field, AI, particularly deep learning techniques utilizing convolutional neural networks, has garnered considerable attention, particularly for its transformative effects on disease management through analyzing medical images. Currently, there are over 20 FDA-approved AI applications for breast imaging, yet their adoption and utilization vary widely and are generally low. While a substantial portion of the published literature and available AI applications primarily focus on detecting breast cancer using AI, the potential applications of AI in breast imaging extend beyond cancer detection [67]. They include risk assessment, breast density quantification, workflow optimization, triage, quality assessment, response evaluation to neoadjuvant therapy, and image enhancement.

Although scientific evidence confirms the diagnostic efficiency of artificial intelligence in digital mammography cancer detection [26,68], it is understandable for patients to hesitate when envisioning a future without the human presence of the BR. Recent studies have explored patients’ perception of AI in the context of breast imaging [69,70,71], highlighting that the general population does not seem to fully approve of the complete replacement of human figures with computers. Instead, they approve of integrating AI as a supplementary tool to the BR’s judgment [10]. The motivation appears to be relatively simple: the opportunity to interact with a human figure offers the possibility of benefiting from emotional support that a fully automated system may not be able to provide [10]. This emphasizes the importance of a thoughtful and patient-centered approach in implementing advanced technologies, ensuring that the use of AI in clinical practice is effective and respects the individual needs of women involved in managing breast health.

While AI holds promise in enhancing the quality of breast imaging studies through techniques like image enhancement and artifact reduction, it is essential to communicate with patients about the role of AI as a supplementary tool to human expertise rather than a replacement. Patients need assurance that their healthcare physicians are utilizing AI to complement their skills and improve diagnostic accuracy, ultimately enhancing patient care and outcomes.

Effective communication about AI’s role in breast imaging is crucial for fostering patient trust and acceptance of this emerging technology. By transparently communicating AI’s benefits and limitations, healthcare providers can ensure that patients feel empowered and involved in their care, thereby facilitating a patient-centered approach to breast health management. However, further large-scale clinical trials are necessary to validate these emerging techniques and assess their efficacy and cost-effectiveness.

## 9. Conclusions

The landscape of cancer care, particularly in managing BC, underscores the indispensable role of effective communication between healthcare providers, patients, and caregivers.

This is especially crucial given the psychosocial challenges often associated with BC diagnosis and treatment. The integration of clear, empathetic, and culturally sensitive communication strategies, comprehending patients’ emotional needs, and engaging in open dialogue, not only enhances patient satisfaction and treatment adherence, but also contributes to improved overall outcomes and reduced legal complications [2,72,73]. Strategies such as prioritizing communication skills training, utilizing “role-play” technique, communication models like SPIKES, creating supportive environments for consultations, and involving multidisciplinary teams stand as valuable tools for delivering distressing news to patients, emphasizing the importance of providing understandable medical information while addressing emotional needs, and contributing to comprehensive and patient-centered care [8]. Communication skills training programs should receive financial support through unrestricted grants and be endorsed by Professional Societies, with credits awarded for medical education. Furthermore, all outcomes, whether objective or subjective, must be closely aligned with the objectives and content of the courses. To ensure consistency and comparability across studies, validated assessment measures should be employed. Future research should also evaluate the long-term impact of CST, which is essential to assess the maintenance of acquired skills, ultimately contributing to improved patient care and outcomes in breast imaging.

Non-verbal communication cues, such as facial expressions and body language, also play a significant role in fostering patient trust and understanding. Furthermore, it is essential to recognize the impact of illness on caregivers and address their needs as integral members of the patient’s support system. Looking ahead, the integration of artificial intelligence in breast imaging presents both opportunities and challenges. While AI has the potential to enhance diagnostic accuracy and streamline workflows, it is essential to transparently communicate its role as a supplementary tool to human expertise, ensuring patient trust and acceptance. By prioritizing communication skill training and fostering empathetic communication practices, healthcare professionals can better navigate the complexities of BC care, ultimately improving the holistic well-being of patients and their families.

Additionally, implementing dedicated guidelines on the involvement of psycho-oncologists, for example, the guidelines published by Gilligan et al. [74], and allocating resources for psychological support to cancer patients while receiving bad news might facilitate BRs and improve good quality and comprehensive cancer care for patients [8,53].

In conclusion, effective communication in BC patient management involves a multifaceted approach that considers patients’ emotional needs, cultural backgrounds, and the evolving landscape of medical technology. By prioritizing patient-centered communication, healthcare professionals can optimize outcomes and enhance the overall quality of care for BC patients and their caregivers.

## Figures and Tables

**Figure 1 healthcare-12-01145-f001:**
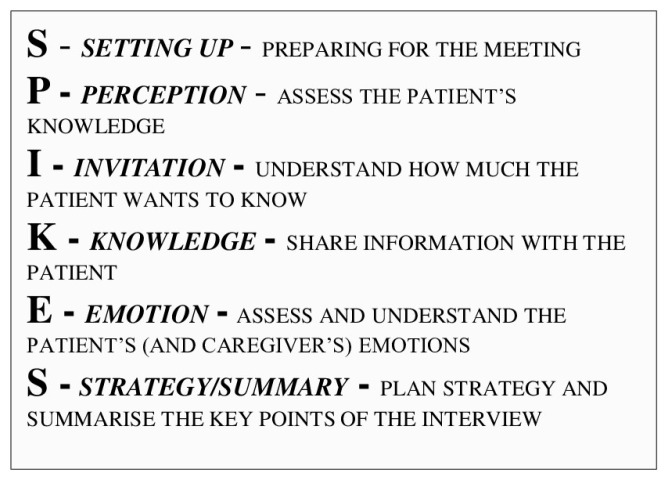
The SPIKES protocol.

## Data Availability

Not applicable.

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
