# Peer review of "Radiologists’ Communicative Role in Breast Cancer Patient Management: Beyond Diagnosis"

_healthcare, 2024, doi:10.3390/healthcare12111145_

Round 1
Reviewer 1 Report
Comments and Suggestions for Authors
Dear Authors,
Please find below my suggestions and commentaries:
1. Main Question Addressed by the Research:
The main question addressed could be further refined to explicitly state how communication impacts patient outcomes, interdisciplinary collaboration, and legal ramifications.
2. Originality and Relevance:
While the importance of communication in patient care is well-established, the focus on breast radiologists' communication strategies adds originality to the discussion. However, to enhance originality, the article could develop the specific challenges and opportunities unique to breast cancer patient communication.
3. Contribution to the Subject Area:
By emphasizing the significance of empathetic and comprehensive communication approaches, the article highlights the potential to improve patient satisfaction, treatment adherence, and overall outcomes. To strengthen its contribution, the article should provide more concrete examples or case studies illustrating the application of communication strategies in clinical practice.
4. Methodology Improvements and Further Controls:
While the article provides a comprehensive overview of communication strategies, it lacks empirical evidence or rigorous methodology to support its claims. To enhance credibility, the authors should consider incorporating empirical data or case studies demonstrating the efficacy of communication approaches.
5. Consistency of Conclusions:
The conclusions could be further strengthened by providing specific recommendations for implementing communication strategies in clinical practice. Additionally, addressing potential limitations or challenges in implementing these strategies would enhance the article's credibility.
6. Appropriateness of References:
To ensure comprehensiveness, the authors should consider including a broader range of references, including empirical studies, clinical guidelines, and relevant frameworks or models of communication. Additionally, the introduction could be expanded to offer valuable insights into psychological support and radiological findings as prognostic factors, complementing the themes discussed in the article.
Author Response
- Main Question Addressed by the Research:
The main question addressed could be further refined to explicitly state how communication impacts patient outcomes, interdisciplinary collaboration, and legal ramifications.
Thank you. We have revised the text to better highlight the underlined aspects. We added these concepts at the end of the special considerations paragraph.
- Originality and Relevance:
While the importance of communication in patient care is well-established, the focus on breast radiologists' communication strategies adds originality to the discussion. However, to enhance originality, the article could develop the specific challenges and opportunities unique to breast cancer patient communication.
we have added a paragraph on the importance of the role of the radiologist in communicating with the patient with breast cancer.
- Contribution to the Subject Area:
By emphasizing the significance of empathetic and comprehensive communication approaches, the article highlights the potential to improve patient satisfaction, treatment adherence, and overall outcomes. To strengthen its contribution, the article should provide more concrete examples or case studies illustrating the application of communication strategies in clinical practice.
we have added a paragraph as per your request.
- Methodology Improvements and Further Controls:
While the article provides a comprehensive overview of communication strategies, it lacks empirical evidence or rigorous methodology to support its claims. To enhance credibility, the authors should consider incorporating empirical data or case studies demonstrating the efficacy of communication approaches.
We have revised the text according to your suggestions. Since the article is a review, we have not included any patient-based case studies.
- Consistency of Conclusions:
The conclusions could be further strengthened by providing specific recommendations for implementing communication strategies in clinical practice. Additionally, addressing potential limitations or challenges in implementing these strategies would enhance the article's credibility.
We have revised the conclusion according to your requests.
- Appropriateness of References:
To ensure comprehensiveness, the authors should consider including a broader range of references, including empirical studies, clinical guidelines, and relevant frameworks or models of communication. Additionally, the introduction could be expanded to offer valuable insights into psychological support and radiological findings as prognostic factors, complementing the themes discussed in the article.
We have updated the introduction and references as per your request.
Reviewer 2 Report
Comments and Suggestions for Authors
The study investigates the communication skills adopted by breast radiologists. Specifically, the study examines the ramifications of communication on patient experience, interdisciplinary collaboration, and legal ramifications utilizing a SPIKES protocol. Based on the study, the researchers advocate for implementing guidelines pertaining to psychological support and the allocation of sufficient resources to ensure the provision of holistic and patient-centered cancer care. Overall, it is a well-thought-out paper, but I have some suggestions that I would like the authors to consider/address to improve the quality of the manuscript.
· In lines 127-138, the authors discuss a lot of different scenarios and highlight what would be good practices. Is this based on research? I ask because I don’t see any citations/references. While logically, it makes sense that it would be interesting to validate through prior studies.
· I see this as a review paper, but I don’t see the methodology used by the authors to conduct the review.
· While authors bring up a lot of interesting points and considerations that BR should keep in mind, I am unsure how authors developed these “recommendations” or potential “policies.”
· In the author contribution statements, the authors mention AN developed the methodology. Can you please add more details regarding this to the paper? Also, what do you mean by “validation” in the author contributions?
Comments on the Quality of English LanguageMinor typos and grammatical changes. No major issues.
Author Response
The study investigates the communication skills adopted by breast radiologists. Specifically, the study examines the ramifications of communication on patient experience, interdisciplinary collaboration, and legal ramifications utilizing a SPIKES protocol. Based on the study, the researchers advocate for implementing guidelines pertaining to psychological support and the allocation of sufficient resources to ensure the provision of holistic and patient-centered cancer care. Overall, it is a well-thought-out paper, but I have some suggestions that I would like the authors to consider/address to improve the quality of the manuscript.
- In lines 127-138, the authors discuss a lot of different scenarios and highlight what would be good practices. Is this based on research? I ask because I don’t see any citations/references. While logically, it makes sense that it would be interesting to validate through prior studies.
Thank you. We have adjusted the number of references according to your specific request.
- I see this as a review paper, but I don’t see the methodology used by the authors to conduct the review.
The article (as already highlighted at the beginning of the submission to the academic editor) has been written according to the guidelines: https://guides.mclibrary.duke.edu/sysreview/types, which have been accepted by the academic editor.
The article belongs to the state-of-the-art review category, and all formal rules of the proposed guidelines for this type of review have been followed.
- While authors bring up a lot of interesting points and considerations that BR should keep in mind, I am unsure how authors developed these “recommendations” or potential “policies.”
We have included clear recommendations in the conclusions.
- In the author contribution statements, the authors mention AN developed the methodology. Can you please add more details regarding this to the paper? Also, what do you mean by “validation” in the author contributions?
The contribution statement was written in accordance with the ICMJE criteria, which include the term validation. (https://www.icmje.org/icmje-recommendations.pdf)
Reviewer 3 Report
Comments and Suggestions for Authors
Dear authors, I do not see the scientific contribution of your work.
Well, It is indeed a review paper, but even in such a form it is necessary to present the method of selection of cited previous studies, as well as to present some tables (e.g.) of the way organizing and analysis them...
Author Response
Dear authors, I do not see the scientific contribution of your work.
Well, It is indeed a review paper, but even in such a form it is necessary to present the method of selection of cited previous studies, as well as to present some tables (e.g.) of the way organizing and analysis them...
We regret the reviewer's statement.
We have tried to improve the text by following the methodology proposed by the guidelines https://guides.mclibrary.duke.edu/sysreview/types
Round 2
Reviewer 1 Report
Comments and Suggestions for Authors
In the current form I consider the manuscript acceptable for publication.
Reviewer 3 Report
Comments and Suggestions for Authors
Beyond Diagnosis